# Presentation and Prognosis of Primary Expansile and Infiltrative Mucinous Carcinomas of the Ovary

**DOI:** 10.3390/jcm11206120

**Published:** 2022-10-17

**Authors:** Marine Huin, Jerome Lorenzini, Flavie Arbion, Xavier Carcopino, Cyril Touboul, Yohann Dabi, Yohan Kerbage, Hélène Costaz, Lise Lecointre, Vincent Lavoué, Pierre-Adrien Bolze, Cyrille Huchon, Alexandre Bricou, Geoffroy Canlorbe, Camille Mimoun, Sofiane Bendifallah, Tristan Gauthier, Gilles Body, Lobna Ouldamer

**Affiliations:** 1Department of Gynecology, Tours University Hospital, 37044 Tours, France; 2INSERM U1069, Université François-Rabelais, 37044 Tours, France; 3Department of Pathology, Tours University Hospital, 37044 Tours, France; 4Department of Obstetrics and Gynecology, Hôpital Nord (AP-HM), Aix-Marseille University (AMU), 13397 Marseille, France; 5Department of Obstetrics and Gynecology, Centre Hospitalier Intercommunal, 94000 Créteil, France; 6Department of Gynecologic Surgery, Jeanne de Flandre Hospital, Centre Hospitalier Universitaire de Lille (CHRU LILLE), Rue Eugene Avinée, 59037 Lille, France; 7Department of Surgical Oncology, Georges-Francois Leclerc Cancer Centre, 21000 Dijon, France; 8Department of Surgical Gynecology, Strasbourg University Hospital, 67000 Strasbourg, France; 9Department of Gynecology, INSERM 1242, COSS, Rennes University Hospital France, Université de Rennes 1, 35042 Rennes, France; 10Department of Gynecologic and Oncologic Surgery and Obstetrics, Lyon Sud University Hospital, Hospices Civils de Lyon, Université Lyon 1, 69008 Lyon, France; 11Department of Gynecology, CHI Poissy-St-Germain, EA 7285 Risques Cliniques et Sécurité en Santé des Femmes, Université Versailles-Saint-Quentin en Yvelines, 78000 Versailles, France; 12Department of Gynecology, Jean-Verdier Hospital—AP-HP, Bobigny University, 93140 Bondy, France; 13Department of Gynecologic and Breast Surgery and Oncology, Pitié-Salpêtrière University Hospital—AP-HP, 75013 Paris, France; 14Department of Gynecology and Obstetrics, Lariboisiere Hospital, 750019 Paris, France; 15Department of Gynecology and Obstetrics, Tenon University Hospital, Assistance Publique des Hôpitaux de Paris (AP-HP), 75020 Paris, France; 16Department of Gynecology and Obstetrics, Hôpital Mère-Enfant, CHU Limoges, 8 Avenue Dominique Larrey, 87042 Limoges, France

**Keywords:** expansile mucinous carcinomas, mucinous ovarian carcinoma, lymphadenectomy, prognosis

## Abstract

**Objective:** The aim of the present study was to evaluate evolution and prognosis of mucinous ovarian carcinomas (mOC), with respect to the two invasive patterns: expansile and infiltrative invasion. **Methods:** This was a descriptive, retrospective, multicenter study conducted in 13 French centres from 1 January 2001 to 31 December 2019. All patients operated on for epithelial ovarian neoplasia of the mucinous type (infiltrative/expansile) were included, whether the surgery was performed immediately or after neoadjuvant chemotherapy. **Results:** A total of 94 women with mucinous carcinomas were included in the present study. Mucinous tumours were divided into 35 expansile (37%) and 59 infiltrative (63%) mOC. There was a statistically significant difference in early and late stages at initial diagnosis between expansile and infiltrative mOC. None of the expansile mOC showed metastatic lymph nodes, whereas almost a quarter of the infiltrative mOC were metastatic to the pelvic/para-aortic region. There was a clear difference in RFS, in favour of expansile mOC, with 90% survival at 5 years, compared with 60% for infiltrative mOC. **Conclusions:** Although infiltrative and expansile mOC belong to the same histological family, they present many distinctions in clinical presentation, histological invasion, and disease course.

## 1. Introduction

Mucinous ovarian carcinomas (mOC) are invasive mucinous neoplasms composed of gastrointestinal-type cells. They belong to the family of epithelial tumours of the ovary but differ from the rest of the subcategories of this family, particularly serous tumours.

Indeed, it is a rare subcategory of ovarian tumours, with an estimated incidence of 3% of all epithelial ovarian cancers [1], with only 20% of primary ovarian mucinous tumours, compared with 80% of metastatic mucinous tumours [2]. Clinically, they are mostly presented as large multiloculated cystic tumours, more often unilateral [3], and they mainly affect a population of young [4,5], non-white [6] women at earlier stages [2]. They have a better prognosis in the early stages, but a poorer prognosis than other types of epithelial cancers in the advanced stages [7].

Recently, in Lee and Shully’s classification, malignant mucinous cancers have been separated into two categories, according to the size and invasion patterns of the pathology: expansile carcinomas and infiltrative carcinomas. The two patterns may coexist; however, the expansile pattern is more common. Expansile mucinous carcinomas have no stromal destruction and are more or less associated with stromal hypertrophy of 10 mm^2^ or more than 3 mm in two dimensions. In contrast, infiltrative mucinous carcinomas show glandular invasion, cluster cells with inconsistent stromal infiltration [2]. In 2014, the World Health Organization (WHO) also adopted Lee and Schully’s classification for mucinous carcinomas of the ovary. In 2020, the WHO classification defined microinvasion as measuring less than 5 mm in the greatest linear extent. The mOC were divided in expansile/confluent carcinomas (Figure 1A) and infiltrative/destructive carcinomas (Figure 1B), each measuring at least 5 mm in the linear extent [8]. An infiltrative pattern, particularly in the setting of bilateral ovarian involvement, should raise suspicion for metastatic mucinous carcinoma and evaluation for an extraovarian source is mandatory.

Histologically speaking, there is similarity in the degree of nuclear atypia when lesions are confined to the ovary between mOC with expansile invasion with mucinous borderline ovarian tumours (mBOT). mBOT are more common than primary mOC. Moreover, they can coexist, as there is often a continuum of architectural and cytological atypia that includes benign, borderline, and carcinomatous areas.

Histological criteria are critical in making the correct diagnosis and classification systems have been area of debates making epidemiological conclusions difficult, with the majority of clinical studies investigating epithelial ovarian cancer as a whole.

The aim of the present study was to evaluate evolution and prognosis of mOC, with respect to the two invasive patterns: expansile and infiltrative invasion.

## 2. Material and Methods

### 2.1. Study Design and Study Population

This was a descriptive, retrospective, multicenter study conducted in 13 French centres Centre Hospitalier Régional Universitaire (CHRU) de Tours, Tenon University Hospital, CHRU de Marseille, Centre Anti-cancéreux de Dijon, Centre Hospitalier Universitaire (CHU) de Lyon Sud, CHRU de Lille, Hôpital de La Pitié Salpêtrière, Centre Hospitalier Intercommunal de Créteil, CHRU de Rennes, Hôpital Lariboisière, CHU Jean Verdier, Centre Hospitalier Intercommunal de Poissy/Saint-Germain-en-Laye, CHU de Limoges, and CHU de Strasbourg from 1 January 2001 to 31 December 2019. The Ethics committee for research in Obstetrics and Gynecology approved the research protocol (CEROG 2019-GYN-605).

### 2.2. Inclusion Criteria

All patients operated on for a primary epithelial ovarian neoplasia of the mucinous type (infiltrative/expansile) were included, whether the surgery was performed immediately or after neoadjuvant chemotherapy (NACT). Histologists from the Rare Malignant Tumors of the Ovary (TMRO) network examined the hematoxylin and eosin (H&E) stained histopathology slides of the available cases.

The histotypes of mOC were redefined by the 2020 WHO classification [8].

### 2.3. Exclusion Criteria

Exclusion criteria were lack of sufficient data to use of the patient files.

The co-existence of non-mucinous ovarian carcinoma.

Ovarian metastases from a gastrointestinal primary.

### 2.4. Variables and Measures

For each patient, the following variables were collected:

-Intrinsic criteria for each patient: age, parity, and body mass index (BMI, which is weight divided by the square of height, expressed in kg/m^2^. We used the WHO classification of BMI. Thus, underweight was defined by a BMI <18.5 kg/m^2^, normal weight: 18.5 < BMI < 25 kg/m^2^, overweight: 25 < BMI < 30 kg/m^2^, and obesity by a BMI > 30 kg/m^2^).-Presence of type 1 or type 2 diabetes, presence of high blood pressure (HBP), menopausal status and use of menopausal hormone replacement therapy (HRT), smoking status.-Assessment of potential genetic predisposition: personal and family history of breast, endometrial, colon, or ovarian cancer. A cancer predisposition mutation was also recorded.

(1)At diagnosis:
-ASA anaesthetic score corresponding to (1) normal patient, (2) patient with moderate systemic abnormality, (3) patient with severe systemic abnormality, (4) patient with severe systemic abnormality representing a constant life threat, (5) moribund patient, and (6) patient declared brain dead; weight loss at diagnosis (in kg).-For biological tests: serum CA 125 level expressed as IU/mL (N < 35 IU/mL), serum CA 19.9 level expressed as IU/mL (N < 37 IU/mL).-Imaging data at diagnosis.(2)For initial stage assessment:
-The type of surgery that resulted in a histological diagnosis: laparoscopy alone or combined with concomitant; delayed laparotomy or upfront laparotomy.-Initial stage of disease according to the 2014 International Federation of Gynecology and Obstetrics (FIGO) and TNM classification (7th edition).

The treatment sequence for each patient was decided in a multidisciplinary consultation meeting after evaluation of the patient criteria and tumour characteristics. NACT usually consisted of three or four courses of carboplatin plus paclitaxel, with imaging reassessment (PET/CT or CT alone). Surgery, performed within four weeks after the end of NACT, consisted of hysterectomy associated with bilateral salpingo-oophorectomy, possibly associated with bilateral pelvic and para-aortic lymph node dissection and infragastric omentectomy, if indicated. Additional procedures were performed in case of macroscopically visible, residual, and resectable disease. Residual disease was assessed at the end of the surgery between no residual disease (R0), residual disease <1 cm (R1), and residual disease >1 cm (R2). After the final histological results were obtained, each patient’s case was discussed again at the multidisciplinary consultation meeting and adjuvant treatments were proposed (chemotherapy-associated, if indicated with a targeted treatment by angiogenesis inhibitor (anti-VEGF), namely bevacizumab, or a targeted treatment by PARP inhibitor (Olaparib), according to the patient’s BRCA status).

In terms of therapy, the following variables were collected:(a)In case of NACT
-The presence of NACT, if applicable, the number of courses before surgery (if applicable) and the different chemotherapy regimens used.-For biological examinations: serum CA 125 level expressed in IU/mL (N < 35 IU/mL), serum CA 19.9 level expressed in IU/mL (N < 37 IU/mL), after three courses.-For imaging: a CT scan and/or PET-CT scan after three courses of NAC. with the observed involvement and associated RECIST criteria.(b)In case of primary cytoreductive surgery (CRS) or interval CRS
-The type of surgery among primary CRS or interval CRS.-Surgical procedures among right and/or left adnexectomy, total hysterectomy, infundibulopelvic ligament removal, infra-gastric or infra-colic omentectomy.-Lymph node procedures among: lombo-aortic lymphadenectomy, bilateral pelvic lymphadenectomy, hepatic hilum lymphadenectomy.-Peritoneal procedures among biopsies or removal of the rectouterine pouch, of the pre-vesical peritoneum, of the right and/or left parieto-colic gutter, resection of the right and/or left diaphragmatic peritoneum, as well as their surface estimated in cm^2^.-Digestive procedures among appendectomy, cholecystectomy, recto-sigmoidectomy, right or left colectomy and/or transverse colectomy, bowel resection, splenectomy, hepatic nodule resection, Glisson’s capsule resection, falciform ligament resection, partial gastrectomy.-Urinary procedures including partial or total cystectomy.-Other procedures among diaphragmatic resection, fulguration procedures with electric energy, nodule resection (peritoneum, mesentery, mesocolon).-Bypass or protective procedures among digestive anastomoses, stomas.-The presence of drains among pleural drain, abdominal drain, nasogastric tube.(c)In case of adjuvant chemotherapy
-The number of courses after surgery and the different lines of chemotherapy used.-On the histological reports:-The tumour size expressed in millimetres.-The histological type of the lesion and its invasion type. Currently, there is no standardized grading system for primary mOC, according to the recommendations of the 2020 World Health Organization classification. The expansile invasive pattern displays marked glandular crowding, with little or absent intervening stroma, creating a labyrinth appearance. Papillary and cribriform areas may be present.

The infiltrative pattern is characterized by irregular glands, nests, and single cells with malignant cytological features, often in a desmoplastic stroma.

-Tumour extension on the different surgical specimens (ovaries, uterus, tubes, omentum, peritoneal resections, digestive and urinary resections…).-Immunohistochemical markers identified.-Presence of lympho-vascular space involvement (LVSI—defined as the presence of tumour cells within the lymphatic or vascular capillaries draining the primary tumour).-The number of nodes removed, and the number of positive lymph nodes.-The presence of an associated other histological contingent.-The presence of an associated borderline contingent.

In order to assess prognosis and survival, the existence of any known loco-regional or metastatic recurrence at the date of data collection (December 2020) was sought by specifying the date of diagnosis, the date of the last news and the date of death of the patients.

### 2.5. Statistical Analysis

Baseline demographic and clinical characteristics were summarized by continuous and categorical variables. Categorical variables were compared with the Chi^2^ [2] test or Fisher’s exact test. Differences between continuous variables were analysed with the Student’s *t*-test or the Mann–Whitney test. A value of *p* < 0.05 was considered statistically significant.

Patient survival was calculated in univariate analysis by the log-rank test and in multivariate analysis by logistic regression (Cox model). Hazard ratios (HR) were reported with their 95% confidence intervals (95% CI). Survival curves were constructed using the Kaplan–Meier method. Survival was calculated as the number of months from the date of ovarian cancer diagnosis to the date of death. Data from patients alive at the time of point were censored.

The data were indexed with an Excel database (Microsoft, redmond, WA, USA), and statistical analyses were performed with R software version 3.1.2 (package Hmisc, Design and Survival Libraries, Vienna, Austria).

## 3. Results

Over the study period, data of a total of 1890 women with ovarian cancers of all histology types were collected. Of these, 94 women with mucinous carcinomas were included in the present study, representing 4.9% of the ovarian cancers in our database. Mucinous tumours were divided into 35 expansile (37%) and 59 infiltrative (63%) carcinomas.

Table 1 and Table 2 show the general characteristics of the whole population and according to histological type, respectively.

The mean age of the patients was 57 years for expansile cancers and 58 years for infiltrative cancers. The BMI was the same (24), but weight loss was noted in 20% (*n* = 12) of the infiltrative carcinomas, compared to 6% (*n* = 2) of the expansile ones.

There were few mutations in both subcategories: 1 case of BRCA2 and KRAS for expansile and 2 cases of BRCA1, 2 cases of Lynch and 1 case of MSH6 for infiltrative type.

Patients with expansile mOC were more smokers (20% vs. 5%) and had more metabolic syndrome (34% vs. 22%) than invasive mucinous carcinomas.

Regarding the clinical presentation, invasive carcinomas induced ascites production in 75% of cases, whereas the presence of ascites was found in only 17% of expansile carcinomas.

Surgical management, either therapeutic or diagnostic, was performed for all patients in our population. There was a statistically significant difference in the number of primary surgeries (*p* < 0.0001) between expansile and invasive mOC. Indeed, all cases of expansile mOC were able to benefit from CRS, with 97% (*n* = 34) undergoing primary CRS and only one interval CRS. Conversely, 83% (*n* = 49) of the infiltrative mOC could be operated on for curative purposes, divided into 52% (*n* = 31) primary CRS, 24% (*n* = 14) interval CRS and 8% (*n* = 5) closure surgery. Table 3 shows the histological characteristics, according to the type of tumour, expansile or infiltrative. In the majority of cases, CRS was complete for the expansile (88%) and infiltrative (80%) groups.

Only one case of LVSI was found in the expansile group, while nine cases of LVSI were found in the infiltrative group (15% of patients).

Bilateral ovarian and capsular involvement was found in 20% of patients (*n* = 12) with infiltrative mOC, compared to 6% of patients with expansile mOC (statistically significant difference *p* < 0.01). Of the patients operated on for infiltrative carcinoma (*n* = 48), 30 underwent para-aortic lymphadenectomy (62%) with 18% (*n* = 9) of lymph node invasion on pathological examination. Pelvic lymphadenectomy was also performed in 33 patients (68%) operated on for invasive mOC with 12% (*n* = 7) lymph node involvement.

Conversely, 31% (*n* = 11) of the patients operated on for expansile mOC had a para-aortic lymphadenectomy, and 31% (*n* = 11) a pelvic lymphadenectomy. No lymph node invasion was found in the pelvic or para-aortic areas.

There was also a statistically significant difference in FIGO stage between expansile and invasive mOC with 83% (*n* = 29) of expansile mOC having an early stage at diagnosis, compared to 32% (*n* = 19) of infiltrative mOC, i.e., 51% early stage in the whole population.

Similarly, 31% (*n* = 11) of the expansile mOC had a normal CA15, compared to 8% (*n* = 5) of the infiltrative mOC.

Regarding chemotherapy treatments, only one patient in the expansile carcinoma group had NACT versus 15 patients (25%) in the infiltrative carcinoma group. Adjuvant chemotherapy was given to 20% of patients in the expansile carcinoma group (*n* = 7) vs. 46% of patients in the infiltrative carcinoma group (*n* = 27).

A total of 25% of patients with invasive carcinoma recurred (*n* = 15), and 4 patients had metastatic disease, mainly lymph node disease with peritoneal carcinomatosis. Only two patients in the expansile carcinoma group recurred (6%), and one presented a pulmonary metastasis. This difference in recurrence between infiltrative and expansile mOC was also statistically significant (*p* < 0.03). Figure 1 shows the overall survival (OS) of the whole population. It shows a decrease in survival of 20% of the patients in 48 months and then a stability of survival. Figure 2, illustrating recurrence-free survival (RFS) according to histological type, shows a clear difference between our two population categories: 90% of patients with expansile mOC have a RFS at 5 years, which remains stable. Conversely, the RFS of patients with invasive mOC decreases progressively without stabilisation, with 60% survival at 5 years.

## 4. Discussion

Our study compared the profiles and evolution of infiltrative and expansile mOC. We were able to identify two entities with different prognosis. Expansile mOC were clearly less aggressive, with almost three quarters of early stages at diagnosis, primary CRS in 97% of cases, rare LVSI, and few recurrences. In contrast, infiltrative mOC were more likely to be advanced in stage, with primary CRS possible in 50% of cases, greater lymph node involvement, and a greater number of recurrences with metastatic progression.

In our study, there was a statistically significant difference in early and late stages at initial diagnosis between expansile and infiltrative mOC. The study by Muyldermans et al. [9] was in agreement with our results, with 21 out of 23 expansile mOC diagnosed at stage 1 without lymph node involvement versus 12 out of 21 infiltrative mOC stage 1 carcinomas with 3 lymph node involvement (out of 10 operated patients).

When these two categories of mOC were studied together, a large number of early stages (51% of patients) were found at diagnosis. This is in agreement with Schiovane’s study, which compared 4811 mucinous cancers with 40,571 other epithelial ovarian cancers [10] and found that mucinous ovarian carcinomas had an earlier stage (stage 1: 55% vs. 11%; *p* < 0.0001), with a similar 5-year survival for stage 1 between mucinous and other epithelial carcinomas.

Sixty-five % to 80% of mOC are diagnosed at an early stage, according to the FIGO classification. Patients with serous ovarian cancer tend to have an advanced stage, with intraperitoneal spread in over 80% of cases. This difference may be explained by the fact that mOC are usually very large primary tumours (typically >15 cm in diameter) that generate symptoms while the disease is still localized to the ovary.

For decades, mOC was classified as grade 1, 2, or 3, according to the presence or absence of nuclear atypia and the proportion of solid glandular component. However, in 2014, the World Health Organization (WHO) introduced a new diagnostic classification of mOC, with two categories based on growth pattern: expansile (confluent) and infiltrative subtype. The 2020 WHO classification modified definition of invasion (5 mm in greater linear extent or more versus 10 mm^2^ in area or over 3 mm in each of the two linear dimensions). The distinction between expansile and infiltrative subtypes is clinically important in stage I disease. The expansile growth pattern suggests a lower metastatic potential, and several studies, although small, have confirmed that the risk of relapse in women with stage I expansile mucinous ovarian cancer is extremely low. In contrast, infiltrative mOC is more aggressive, with at least 26% of women having more advanced, non-localized disease at diagnosis; in 17–30% of patients who appear to have stage I disease, lymph node metastases are detected. Even if the cancer is diagnosed at an early stage, the prognosis for women with mucinous infiltrative ovarian cancer is much poorer, with fatal relapse reported in 15–30% of patients with stage I disease.

Regarding therapeutic management, for all patients in our study, surgery was prioritized as first line and resulted in a large number of CRS with no macroscopic residue. An appendectomy should be performed systematically, as it allows for the diagnosis of 4% of metastases, despite a macroscopically healthy appendix [11].

Another debate persists concerning the indication for staging with lymph node dissection in malignant mucinous ovarian tumours. Several studies have found no indication for pelvic and para-aortic lymphadenectomy in mOC [12,13]. However, these studies did not differentiate between expansile and infiltrative mOC. In our study, none of the expansile mOC showed metastatic lymph nodes, whereas almost a quarter of the infiltrative mOC were metastatic to the pelvic/para-aortic region.

More recently, several studies looking separately at expansile and infiltrative mOC have found similar results [8,13]. In the study of Gouy et al. [14], none of the lymph node samples were metastatic in expansile mOC unlike infiltrative mOC in the early stages, but an interest in peritoneal staging was reported for all mucinous carcinomas. It should be noted that the therapeutic value of lymphadenectomy remains to be demonstrated [15]. Thus, in 2019, a French recommendation [16,17] was issued (in agreement with FRANCOGYN, CNGOF, SFOG, GINECO-ARCAGY, and INCA) recommendations on the absence of indication for lymphadenectomy in expansile mOC.

Concerning adjuvant treatments, several studies agree on the existence of chemo-resistance to platinum drugs in mucinous carcinomas [2,18], despite the fact that, in the Nasioudis study [19], the use of chemotherapy in advanced stages was associated with better survival (HR 0.63, 95% CI 0.55, 0.72). These results are in agreement with our study, in which 46% of patients with invasive mOC received NACT and 25% adjuvant chemotherapy. However, the cases of infiltrative mOC were of a more advanced stage.

Furthermore, in our study, there was a clear difference in RFS, in favour of expansile mOC with 90% survival at 5 years, compared with 60% for infiltrative mOC. Genestie et al. [20] also found a decrease in OS (*p* < 0.0024) and RFS (*p* < 0.0060) for infiltrative mOC, compared with borderline and expansile mOC. Similarly, in the multivariate study by Hada et al. [21], infiltration was associated with a worse prognosis in RFS (HR 9.01, *p* < 0.01) and OS (HR 17.56, *p* < 0.01).

Furthermore, several studies agree on the worse [22] and significantly poorer [7] prognosis of advanced mOC, compared to serous carcinomas.

Overall survival is higher for the majority of patients with stage I disease than for those with non-mucinous histological subtypes (hazard ratio, 0.52; 95% confidence interval (CI), 0.30 to 0.92). However, the trend is reversed for women with stage III or IV mOC, who have significantly poorer OS than women with non-mucinous histological subtypes (hazard ratio, 2.81; 95% CI, 2.47 to 3.21).

Mucinous cancer has a distinct clinical presentation and course. In all histological classifications, most women with malignant ovarian carcinoma will be diagnosed at an advanced stage of the disease. Although this is typical of serous adenocarcinomas, most women with mOC will, in fact, be in stage I, where the 5-year survival rates are excellent (nearly 90%). Unfortunately, the same cannot be said for advanced disease. Hess et al. showed that women with stage III/IV mOC had a PFS of only 5.7 months, compared with 14.1 months for other epithelial histologies. Overall survival was also significantly shorter (12.0 months versus 36.7 months) [23]. This was confirmed by a further review of six phase III trials from the Gynecologic Oncology Group in women with stage III epithelial ovarian carcinoma receiving adjuvant chemotherapy; women with mOC had a median OS of only 14.8 months, compared with 45.2 months for women with serous carcinoma [24].

## 5. Conclusions

Although infiltrative and expansile mOC belong to the same histological family, they present many distinctions in clinical presentation, histological invasion, and disease course. Surgical treatment with the achievement of CRS without macroscopic residue is the main curative treatment of mOC.

Invasive mOC are frequently advanced in stages at diagnosis with therapeutic surgical management possible in only 50% of cases. The number of LVSI and lymph node involvement are such that lymph node dissection is still recommended in the latest French guidelines. Adjuvant treatment with chemotherapy is still to be discussed, depending on the stage and the extent of invasion at the final histological examination.

On the other hand, expansile mOC mainly diagnosed at an early stage has a very good prognosis. Moreover, in the absence of infiltrative mOC, lymph node dissection is not recommended. The low number of recurrences does not allow for recommendations on adjuvant chemotherapy to be made.

## Figures and Tables

**Figure 1 jcm-11-06120-f001:**
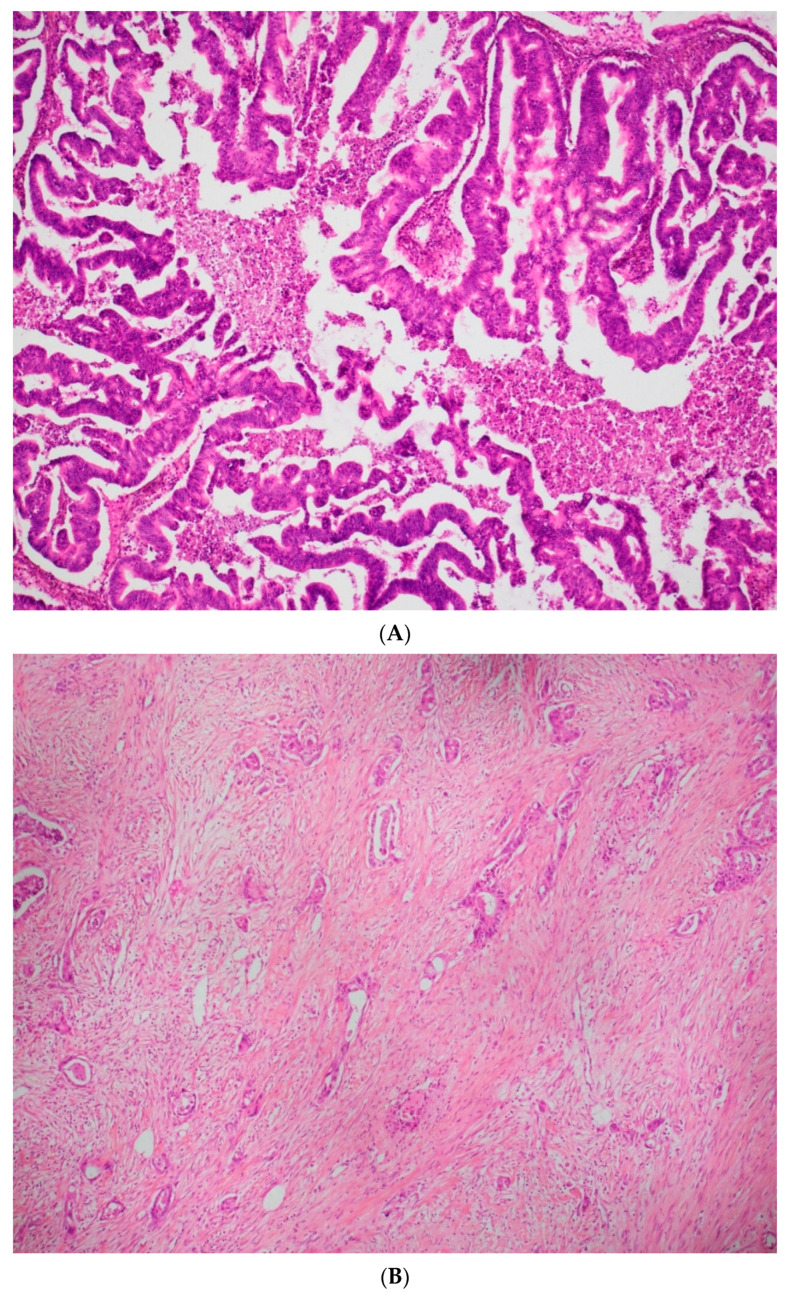
(**A**) ×20 H&E image with expansile/confluent mucinous carcinoma. (**B**) ×20 H&E image with infiltrative/destructive mucinous carcinoma.

**Figure 2 jcm-11-06120-f002:**
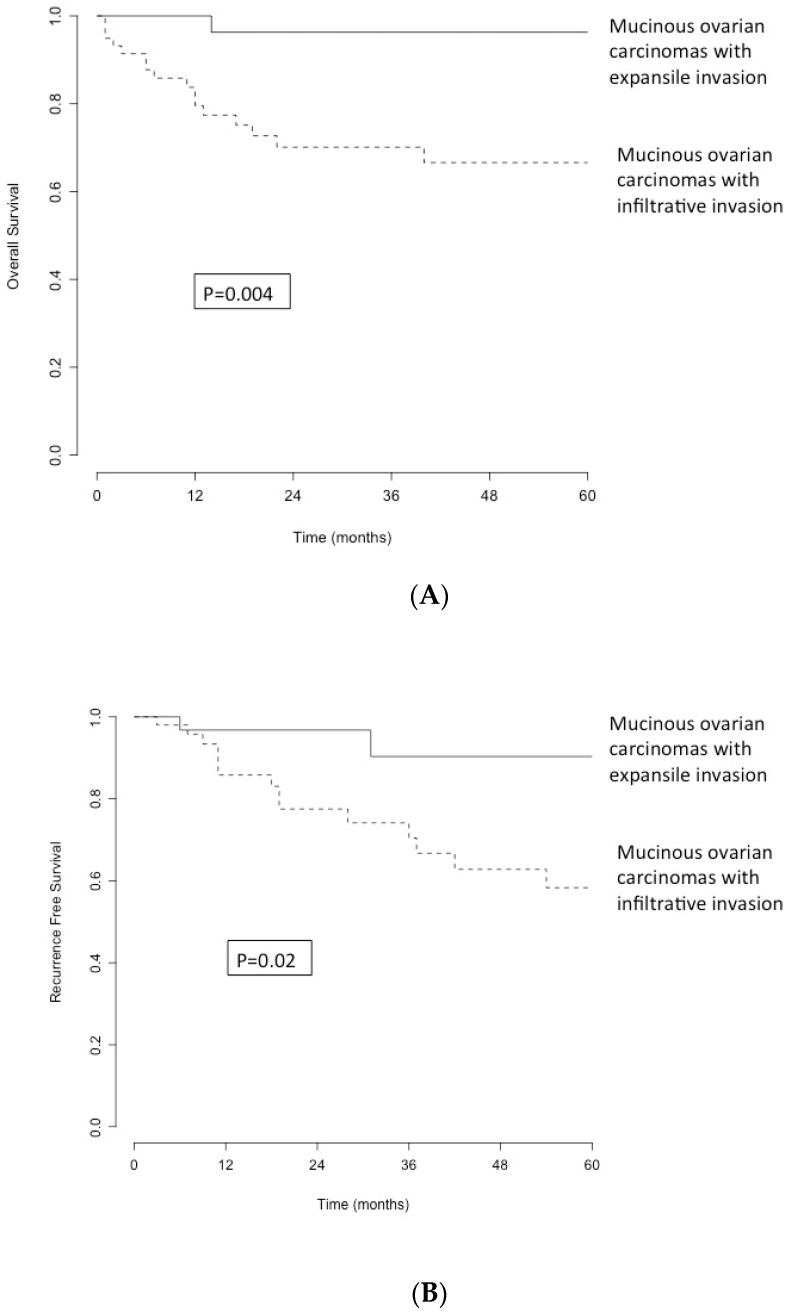
(**A**) Overall survival by histological type (straight line: expansile mOC; dashed line: infiltrative mOC). (**B**) Recurrence-free survival graph by histological type (straight line: expansile mOC; dashed line: infiltrative mOC).

**Table 1 jcm-11-06120-t001:** Demographic characteristics of the population (*n* = 94).

Characteristics	Population (*n* = 94)	NA
Age (years)	56.5 ± 15.4 (18–88)	-
BMI (in kg/m^2^)	25.1 ± 5.9 (14.5–44)	9
Weight loss	14 (15%)	24
Parity		3
0	23	
1	18	
2	28	
3	16	
4	3	
>4	3	
ASA		38
0	0	
1	27 (29%)	
2	22 (23.5%)	
3	7 (7.5%)	
CT at initial diagnosis	56 (60%)	9

Data are presented as mean ± standard deviation (minimum–maximum) or numbers (%); BMI: body mass index/CT: computed tomography/ASA: anaesthetic score/NA: data not available.

**Table 2 jcm-11-06120-t002:** Characteristics of the population according to the type of mucinous tumour (*n* = 94).

	Infiltrative (*n* = 59)	Expansile (*n* = 35)	NA	*p*
Age (median in years)	58	57	0	0.62
Parity (median)	1	2		0.07
Predisposing mutation	5	2		1
FIGO clinical stage at diagnosis			6	<0.0001
Stage I	19	28		
Stage II	0	1		
Stage III	27	3		
Stage IV	9	0		
First surgery	31 (52%)	34 (97%)		<0.0001
Type of surgery			0	<0.0001
Primary cytoreductive surgery	30 (51%)	34 (97%)		
Interval cytoreductive surgery	14 (24%)	1 (3%)		
Closure cytoreductive surgery	5(8%)	0		
Surgery exploration	10(17%)	0		
Residual disease at end of surgery			2	0.14
R0 (complete surgery)	40 (83%)	31 (91%)		
R1 (optimal surgery)	3 (6.5%)	3 (9%)		
R2 (sub-optimal surgery)	5 (10.5%)	0		
Overall recurrence	15 (25%)	2 (6%)		0.03

Data are presented as numbers (%) or numbers (25th percentile–75th percentile). NA: data not available.

**Table 3 jcm-11-06120-t003:** Histological characteristics according to tumour type (*n* = 93).

	Infiltrative (*n* = 59)	Expansile (*n* = 35)	NA	*p*
Capsule rupture	12 (20%)	2 (6%)		0.01
Lymph node involvement			-	
Pelvic	7 (12%)	0		0.23
Para-aortic	9 (15%)	0		0.10
Immunohistochemical markers				
CK7	22	1		0.86
CK20	9	11		0.03
WT1	6	2		0.87
Estrogen receptors	8	1		0.26
Progesterone receptors	4	0		0.45
P53	5	3		1

Data are presented as numbers (%) or (minimum–maximum) numbers; NA: data not available.

## Data Availability

Not applicable.

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
