# Peer review of "Presentation and Prognosis of Primary Expansile and Infiltrative Mucinous Carcinomas of the Ovary"

_jcm, 2022, doi:10.3390/jcm11206120_

Round 1
Reviewer 1 Report
This is a well-known topic about mucinous ovarian carcinomas. There are many published articles that can be found in the literature about the same topic with a relatively more number of cases. Therefore, the authors must explain what they found new and why they think this manuscript should be published. Moreover, please correct the below points:
1. The introduction is too short. You have to explain well the meaning of expansile mucinous carcinoma and what is the difference with mucinous borderline tumors. Moreover, explain more about the difference of expansile and invasive mucinous carcinomas from the histologic, prognostic, and therapeutic viewpoint.
2. Material and methods section is very unstructured and confusing. I suggest rewriting this section in well-structured paragraphs. Make clear your inclusion and exclusion criteria. Are you sure that ONLY 94 cases of mucinous ovarian carcinoma were diagnosed during 19 years in 13 French Medical Centers? This is a very small number in 19 years of 13 hospitals.
3. Discuss well why your result is important and what is new in your results than reported before.
4. Fig.1 and Fig. 2 are not clear. Make it one figure with A and B sections. Also, all lines (straight and dashed) must be clear in the figures, not only in figure legends.
You have to also add at least one microscopic figure of expansile and invasive mucinous carcinomas.
5. Use the abbreviations appropriately. I don't know what do you exactly mean by using MOC? If it is Mucinous Ovarian Carcinoma, the term is already used in the abstract without abbreviation in the Objective, but you have used MOC in the Results of the abstract for only carcinomas. Again in the introduction, you have used MOC for Malignant mucinous tumours of the ovary.
Author Response
This is a well-known topic about mucinous ovarian carcinomas. There are many published articles that can be found in the literature about the same topic with a relatively more number of cases. Therefore, the authors must explain what they found new and why they think this manuscript should be published. Moreover, please correct the below points:
- The introduction is too short. You have to explain well the meaning of expansile mucinous carcinoma and what is the difference with mucinous borderline tumors. Moreover, explain more about the difference of expansile and invasive mucinous carcinomas from the histologic, prognostic, and therapeutic viewpoint.
The introduction was rewrited as suggested by the reviewer
- Material and methods section is very unstructured and confusing. I suggest rewriting this section in well-structured paragraphs. Make clear your inclusion and exclusion criteria.
Modified as suggested by the reviewers
Are you sure that ONLY 94 cases of mucinous ovarian carcinoma were diagnosed during 19 years in 13 French Medical Centers? This is a very small number in 19 years of 13 hospitals.
We only included cases that were centrally reviewed by the histologists fromthe Rare Malignant Tumorsof the Ovary (TMRO) network certifiedby the National Cancer Institute (INCa) rare gynecological malignant tumors (Tumeurs Malignes Rares Gynécologiques- TMRG)expert centers.
- Discuss well why your result is important and what is new in your results than reported before.
The most important value of our article is that it is one of the rarest taking into account the new WHO classification
- 1 and Fig. 2 are not clear. Make it one figure with A and B sections. Also, all lines (straight and dashed) must be clear in the figures, not only in figure legends.
Corrected as suggested by the reviewer
You have to also add at least one microscopic figure of expansile and invasive mucinous carcinomas.
Added
- Use the abbreviations appropriately. I don't know what do you exactly mean by using MOC? If it is Mucinous Ovarian Carcinoma, the term is already used in the abstract without abbreviation in the Objective, but you have used MOC in the Results of the abstract for only carcinomas. Again in the introduction, you have used MOC for Malignant mucinous tumours of the ovary.
Corrected

Reviewer 2 Report
Thank you for submitting the manuscript entitled: Presentation and prognosis of primary expansile and invasive mucinous carcinomas of the ovary".
This study present a relatively large cohort of patients with presumably primary mucinous ovarian carcinomas. The conclusions overall support the findings that have been reported by other studies.
However there are the following recommendations:
1. the two type of invasive patterns in ovarian mucinous carcinomas are expansile or infiltrative. The term invasive is a generic term indicating that the carcinoma is invading the ovarian stroma and encompasses bout expansile and infiltrative patterns of invasion. This needs to be updated in the entire manuscript.
2. Not sure what the authors compare mucinous tumors to serous. These are two completely different tumor times, with different etiology, pathogenesis, outcome, prognosis and treatment. It creates unnecessary confusion and inconsistency in the manuscript.
3. Introduction- lines 57-62- please rewrite, redundant English and inaccurate grammar. Lines 63-68- please rewrite definitions according to the 2020 WHO book for tumors of the female genital tract.
4. Materials and Methods: page 4, lines 167-168- grades1-3 are not sued for ovarian mucinous carcinomas. Please update per 2020WHO
5. Materials and Methods: page 4, lines 175 - change to positive lymph nodes not invaded
6. Results: how did the authors make sure all these tumors were primary ovarian mucinous carcinomas and none were metastases for GI primary?
7. survival results between the two patterns of invasion should be compared stage to stage as well
8. was there any difference in survival and outcome between patients with LVI or without?
9. Page 10. please explain all the abbreviations in line 327
Page 10: lines 329-330: please revise English and grammar
Page 10- Lines 343-347- please include reference
10. Conclusions: page 10 - lines 361-363- please revise English for example histological pattern (not invasion)
11. Page 10- line 371- : in the absence of lymph node invasion, lymph node dissection is not recommended"- how is the lymph node invasion determined prior to surgery?
12. Page 10, lines 374-376: this conclusion can not be drawn from this study, please delete
Author Response
Thank you for submitting the manuscript entitled: Presentation and prognosis of primary expansile and invasive mucinous carcinomas of the ovary".
This study present a relatively large cohort of patients with presumably primary mucinous ovarian carcinomas. The conclusions overall support the findings that have been reported by other studies.
However there are the following recommendations:
- the two type of invasive patterns in ovarian mucinous carcinomas are expansile or infiltrative. The term invasive is a generic term indicating that the carcinoma is invading the ovarian stroma and encompasses bout expansile and infiltrative patterns of invasion. This needs to be updated in the entire manuscript.
Modified as suggested by the reviewer
- Not sure what the authors compare mucinous tumors to serous. These are two completely different tumor times, with different etiology, pathogenesis, outcome, prognosis and treatment. It creates unnecessary confusion and inconsistency in the manuscript.
High grade Serous ovarian carcinoma is the most frequent epithelial ovarian carcinoma, and the question of differences of behavior with mOC is a real clinical question
- Introduction- lines 57-62- please rewrite, redundant English and inaccurate grammar. Lines 63-68- please rewrite definitions according to the 2020 WHO book for tumors of the female genital tract.
Modified as suggested by the reviewer
- Materials and Methods: page 4, lines 167-168- grades1-3 are not sued for ovarian mucinous carcinomas. Please update per 2020WHO
Modified as suggested by the reviewer
- Materials and Methods: page 4, lines 175 - change to positive lymph nodes not invaded
Corrected
- Results: how did the authors make sure all these tumors were primary ovarian mucinous carcinomas and none were metastases for GI primary?—
Histologists from the Rare Malignant Tumors of the Ovary (TMRO) network examined the hematoxylin and eosin (H&E) stained histopathology slides of the available cases and patients had underwent investigations to rule out GI primay (colonoscopy, imaging whatever the procedure)
- survival results between the two patterns of invasion should be compared stage to stage as well
not enough patients to compare stage by stage but we modified the overall survival graph to a comparison of the OS of the 2 histological subtypes
- was there any difference in survival and outcome between patients with LVI or without?
Yes and this was a part of another publication only focused on LVSI
- Page 10. please explain all the abbreviations in line 327
modified
Page 10: lines 329-330: please revise English and grammar
corrected
Page 10- Lines 343-347- please include reference
- Conclusions: page 10 - lines 361-363- please revise English for example histological pattern (not invasion)
corrected
- Page 10- line 371- : in the absence of lymph node invasion, lymph node dissection is not recommended"- how is the lymph node invasion determined prior to surgery?
Corrected we meant in the absence of infiltrative mOC, lymph node dissection is not recommended
- Page 10, lines 374-376: this conclusion can not be drawn from this study, please delete
deleted

Round 2
Reviewer 1 Report
Even though the topic is not very interesting, I am satisfied with the authors' revision.